# Cost-efficient boundary-free surface patterning achieves high effective-throughput of time-lapse microscopy experiments

**Guohao Liang**[1◉]**, Hong Yin**[1◉]**, Jun Allard**[2,3]**, Fangyuan Ding**[1,2,4]*

**1** Department of Biomedical Engineering, University of California, Irvine, Irvine, California, United States of America, **2** Department of Mathematics, and Department of Physics and Astronomy, University of California, Irvine, Irvine, California, United States of America, **3** Center for Complex Biological Systems, University of California, Irvine, Irvine, California, United States of America, **4** Center for Synthetic Biology, Department of Developmental and Cell Biology, and Department of Pharmaceutical Sciences, University of California, Irvine, Irvine, California, United States of America

◉ These authors contributed equally to this work.
* dingfy@uci.edu

**Data Availability Statement:** All relevant data are within the paper and its Supporting Information files.

## Abstract

Time-lapse microscopy plays critical roles in the studies of cellular dynamics. However, setting up a time-lapse movie experiments is not only laborious but also with low output, mainly due to the cell-losing problem (i.e., cells moving out of limited field of view), especially in a long-time recording. To overcome this issue, we have designed a cost-efficient way that enables cell patterning on the imaging surfaces without any physical boundaries. Using mouse embryonic stem cells as an example system, we have demonstrated that our boundary-free patterned surface solves the cell-losing problem without disturbing their cellular phenotype. Statistically, the presented system increases the effective-throughput of time-lapse microscopy experiments by an order of magnitude.

## Introduction

Single-cell time-lapse microscopy has been used in many cell studies. It provides a dynamic picture of cellular regulation and has revealed many unexpected cellular regulation mechanisms [1–3] previously hidden with conventional population-based techniques. For instance, dynamic studies have helped discover factors in stem cell fate choices [4, 5], the dynamics of epigenetic regulation [6, 7], and the pulsatile nature of transcription in response to stress [8, 9].

Time-lapse microscopy generally functions in the following way [10, 11]. Live cells are placed under a microscope in a temperature, $CO_2$, and humidity-controlled chamber to allow for long periods of cell dynamics tracking. Images are taken at a set time interval (minutes up to an hour) for an extended amount of time (hours to days) to observe the full process of interest, then all time-series images are assembled to produce a movie. If observation of multiple colonies is desired, a motorized stage could be used to move the culture plate to pre-programmed positions, aligning the objective to the desired colonies at each capture time. In this

**Funding:** This research was funded by the National Institutes of Health (NIH), National Science Foundation DMS 2052668 (to A.J.) and generous startup funds from the University of California-Irvine (to F.D.). F.D. is the recipient of NIH Director's New Innovator Award 1DP2GM149554. The funders had no role in study design, data collection and analysis, decision to publish, or preparation of the manuscript.

**Competing interests:** The authors have declared that no competing interests exist.

plate-scanning mode, images will be assembled based on their respective location and time to produce one movie for each position, providing parallel data output.

However, this modality remains problematic for studies requiring high amplification with extended tracking time and/or large data throughput. For example, entire lineage tracking can take 4–7 days to complete [12, 13] and a given colony of cells needs to be tracked from beginning to end. This is challenging because cells stay mobile throughout the tracking period and may move away from the limited field of view under high amplification, thus voiding the movie at that position. Manual adjustments to the pre-programmed positions are not feasible due to the frequency of image acquisition. Additionally, studies desiring to observe a heterogeneous population (e.g. stem cell heterogeneity [14–16]) are also hindered by this issue, since high throughput is required to sample the wide range of scenarios in a heterogeneous population. In theory, hundreds of positions on a culture dish could be pre-programmed and imaged in the plate-scanning mode, achieving large parallel cell colony input. However, cells randomly moving out of the field of view means that the number of positions retaining cells at the end of recording is unpredictable. Therefore, high effective-throughput, in which most if not all imaging positions retain cells throughout the time-lapse recording, is key for advancing the modality.

Previous works have made many efforts to increase the effective-throughput of time-lapse movies. For instance, wide-field microscopy [17, 18] has been adapted to increase the information density of time-lapse imaging. This system does not take images position-by-position, but instead takes one wide-field image (6 mm X 4 mm) at once. However, this system has lower resolution (700 nm ~ 1.2 μm) [17, 18], rendering it not ideal for experiments requiring high resolution to track cell divisions and movements (such as stem cell differentiation). Additionally, physical boundaries made of PDMS have also been used to confine cells for time-lapse imaging to increase the effective-throughput by minimizing cell loss in lineage experiments [1]. But such methods are not compatible with all types of cell experiment, as the physical confinement and contact induce phenotypic changes [19–21]. Furthermore, photolithography is another approach to deposit either physical or non-physical boundaries for cell confinement [22–24]. However, the high cost of this approach prohibits it from being widely used. Most molecular and cellular biology labs (like us) do not have easy access to clean rooms, which makes the photolithography patterning not only expensive, but also not universally applicable. In addition, using photolithography here can be wasteful, as this technique is capable of nanometer resolution patterning [25], while avoiding cell-loss under the field of view requires only a scale of hundred micrometers patterning.

Here we propose a proof-of-principle high-informative, boundary-free, and cheap cell-patterning method to increase the effective-throughput of time-lapse microscopy. By using PDMS stencils with arrays of pre-punched holes (hundred microns in diameter), we can pattern the glass surface into zones with various adhesion to cells without physical boundaries. This method confines the cultured cells to the high affinity zones at low cell density, but still allows them to grow in the low affinity zones once the high adhesive area is populated. We have demonstrated the system working for sensitive cell types (mouse embryonic stem cell) for a 4-days time-lapse movie. Estimates from simple simulations suggest that our system can improve the throughput by at least an order of magnitude.

## Material and methods

### PDMS stencil with holes

To create isolated spots of cell adhesive surface, we have fabricated a PDMS stencil containing numerous holes (~250 μm diameter). First, we made PDMS using the Sylgard 184 silicone

elastomer kit (10:1 base/curing ratio by weight). Second, a blank PDMS sheet of 250 μm thickness was manufactured by allowing liquid PDMS (degassed by vacuum half hour) to solidify in a metal mold (Fig 1A). The mold was fabricated by a Caltech machine shop (The GALCIT Shop), but it can also be ordered commercially in most off-campus machine shops. Then, the PDMS sheet was transferred onto a thin (~ 1 mm thick) mylar support and placed on the bed of a VLS350 (Universal Laser Systems, Arizona, USA). To produce holes, a pattern was created using the CorelDraw software (Corel, Ottawa, Canada) with a light gray shape (color 0xFDFDFD) where a field of holes is desired. The laser then performed a raster scan at 100% power (S1 Movie), and the scan was repeated three times to ensure the holes fully penetrate the PDMS membrane. Raster scanning was used because it is considerably faster than cutting small holes using the vector mode. Finally, a vector cut was performed to cut along the circular outline of the stencil (S2 Movie), with a notch placed so we can determine the stencil's orientation, as the bottom surface is much smoother and adheres better to glass than the top surface. The stencil was then washed by 95% EtOH to remove vaporized PDMS residue and sterilized.

## Surface patterning

With the fabricated PDMS stencil, we have achieved to coat the glass surface into a cell adhesive pattern. First, we used 95% EtOH to clean the PDMS stencil and applied nitrogen air to blow dry. Second, we put the stencil on top of the glass bottom (Ibidi® 35mm dish 81218–200). One can improve the attachment either by manually squeezing the PDMS onto the glass surface or using a quick vacuum pressure. Third, we performed the plasma treatment, at pressure 800–1000 with medium RF level for 10 mins using the plasma cleaner (PDC-32G,

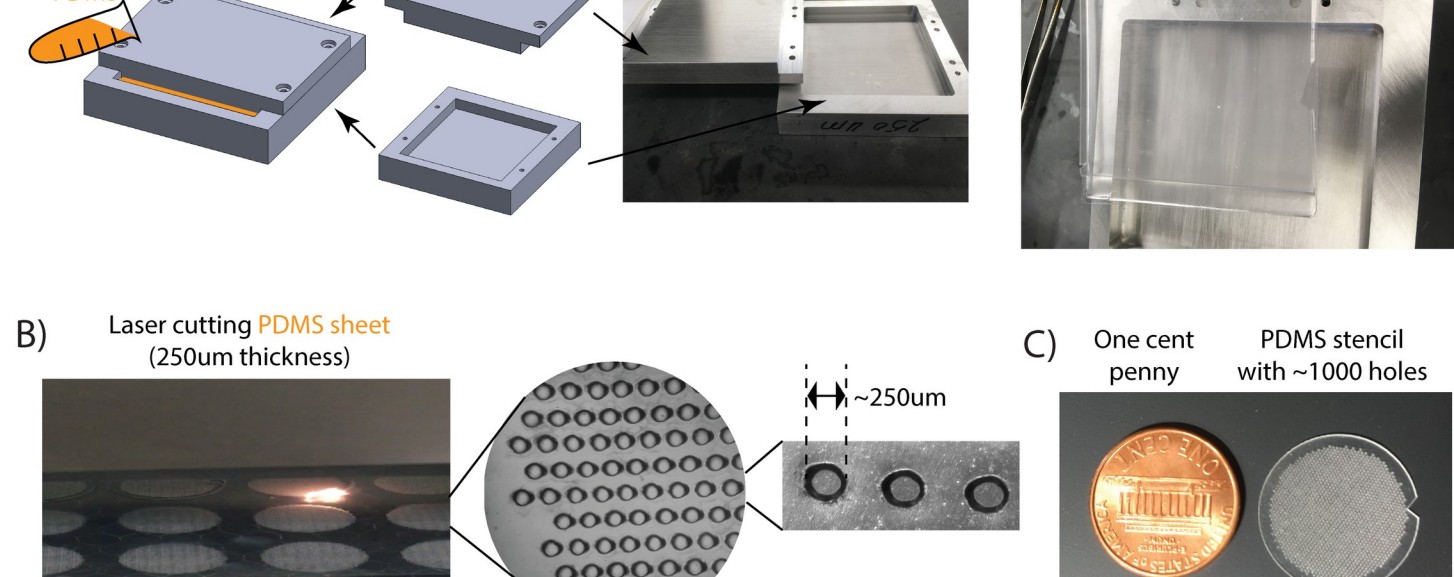

**Fig 1. We have fabricated reusable PDMS stencils by laser cutting.** (A) We used a metal mold, with a 250 μm gap, to manufacture a degassed PDMS sheet of 250 μm thickness (more details described in Materials and Methods). Worth mentioning, PDMS sheets with similar thickness are also commercially available. (B) We used a laser printer to penetrate holes (~250 μm diameters) on the PDMS sheet (more details in Materials and Methods). (C) About 1000 holes were created in a ~19mm diameter area, a size compatible to pattern a 35mm glass-bottom cell culture dish.

HARRICK PLASMA). Fourth, we added 10 ug/ml laminin (human laminin-511, BioLamina) into the dish (about 250ul, the exact volume may vary, as long as it can cover the glass surface and PDMS stencil), with a 5 mins vacuum treatment (to get rid of bubbles in the stencil holes), then incubated at 37˚C for 1 hr. Fifth, we washed the dish with PBS three times, added 0.2% BSA-Biotin solution (Pierce™ Bovine Serum Albumin, Biotinylated Catalog number: 29130), and removed the PDMS stencil. After incubating 1 hr at 37˚C, we washed the dish with PBS three times again. Finally, we added the labeled Streptavidin (Alexa Fluor® 405 conjugate Catalog number: S32351), incubated 10 mins at room temperature, then washed the dish again with PBS. Despite the patterned glass surface capable of being stored in 4C for days, we recommend using a freshly prepared sample for time-lapse movie experiments.

## Culture conditions and cell line construction

Brachyury-GFP mouse embryonic stem cells (E14.1, 129/Ola, from previous publication [26], and we did not test for mycoplasm) are cultured in a humidity controlled chamber at 37˚C with 5% CO2, and plated on tissue culture plates pre-coated with 0.1% gelatin (relatively cheaper compared to laminin, better option for daily culture) and cultured in standard pluripotency-maintaining conditions [27, 28] using DMEM supplemented with 15% FBS (ES qualified, GIBCO), 1 mM sodium pyruvate, 1 unit/ml penicillin, 1 μg/ml streptomycin, 2 mM L-glutamine, 1X MEM non-essential amino acids, 55 mM β-mercaptoethanol, and 1000 Units/ml leukemia inhibitory factor (LIF). Cells were passaged by Accutase (GIBCO). For differentiation experiments, cells were cultured in the same medium, without LIF, but adding 3 μM CHIR.

To label the nucleus with CFP, we have transfected a plasmid with H2B-mCerulean into the Brachyury-GFP E14 cell lines. First, we have cloned H2B-mCerulean into a pcDNA5 expression vector under a PGK promoter. Second, we have followed the standard protocol of FuGENE® HD Transfection Reagent (Promega Corporation) to perform the transfection. Then, the surviving transfected cells with strong CFP signal were sorted by Caltech FACS Facility. Finally, we plated the sorted ~271k cells, let it grow for another 4 days and froze them for further experiments and lab storage.

## Time-lapse microscopy imaging

For the 35 cm glass-bottom dish, we plated about 25k cells with complete cell media to achieve a proper cell density: 1–2 cells per patterned area. The dish was manually swayed [29] to uniformly spread the cells and left in the incubator for 2–3 hr before imaging. Details of time-lapse microscopy setting have been described previously [30]. For each movie, about 100 stage positions were picked manually, and CFP images were acquired every 15 mins with an Olympus 60x oil objective using automated acquisition software (Metamorph, Molecular Devices, San Jose, CA), then differential interference contrast and GFP images were acquired at the beginning and the end of the movie.

## Lineage tracking

Cells were segmented and tracked using the H2B-mCerulean with custom MATLAB code (available upon request), as described in previous publication [6] and detailed again here: (1) Initially, images were processed to correct for inhomogeneous fluorescent illumination by fitting a paraboloid to background (non-cell) pixel intensities, and then normalizing the image by this paraboloid. (2) We then used an integrated segmentation and tracking procedure which combined (a) a pixel-based intensity threshold for segmentation, (b) a tracking algorithm based on global minimization of a cost function that incorporates cell positions and

fluorescence intensities, and (c) heuristics that use discontinuities in tracking to correct segmentation. (3) Finally, all individual cell lineages were checked and corrected manually.

### Simulation of colony growth

Cells are positioned to maintain compactness and the colony is approximately circular. We can implement colony growth in the simulation in two models.

First, in the **synchronized cell cycle** model, we assume all cells have synchronized cell cycles. As shown in S5 Movie, a cell (radius 7.5 μm) was placed at the center of the field of view (1024 pixel by 1024 pixel, with pixel size 216.7 nm, from the value of our microscopy camera). Based on our recorded movies, we set cells moving 1 μm per min and dividing every 12 hr. Every step of a cell moving was in a random direction, and every cell division doubled cell colony size.

Alternatively, if cell cycles are not synchronized, we can simulate **continuous stochastic growth** by employing a Gillespie algorithm [31] in which we track the number of cells n(t) over time. If there are n(t) cells at time t, the mean time to next cell division is

$$t_{next} = -\frac{12\text{hrs}}{n}\ln(\text{rand})$$

where rand is a uniform random number in (0,1). At each division event, the area of the colony is increased by $\pi r_1^2$ here $r_1 = 7.5$ $u$m is the radius of a single cell.

The geometric-mean colony size over 4 days is $\langle r \rangle \approx 120$ μm.

### Analysis of colony movement

We assume the colony migration is well-described by a simple random walk model.

Simple random walks are described by a random walk parameter (diffusion coefficient) $D_{rw}$. Roughly speaking, the random walk parameter $D_{rw}$ is the instantaneous velocity multiplied by the directional persistence length, which is itself roughly the distance the colony migrates before changing direction [32]. The relationship between the mean displacement <r> over a time dt and the random walk parameter is

$$\langle r \rangle = \sqrt{\pi D_{rw} dt}$$

Using $\langle r \rangle = 15$ μm and dt = 15 min gives $D_{rw} = 4.8$ μm$^2$ /min.

### Analytic approximations

We can get a rough estimate under the approximation that the colony size is fixed. For this, we use the geometric mean size $\langle r \rangle \approx 120$ μm. The mean time for a colony of size <r> to have moved to the edge of the adhesive zone [33], where it is approximately 50% escaped, is

$$T = \frac{(R_{zone})^2}{4D_{rw}} = 13.6 \text{ hr}$$

where $R_{zone} = 125$ μm is the radius of the adhesive zone. Note this is a significant approximation since the colony size is varying from much smaller than the adhesive zone during Day 1 to much larger in Day 4.

### Results and discussion

As described in the Materials and Methods and illustrated in Fig 1, we first used the laser to punch holes (about 250 μm diameter) on a PDMS sheet, which took less than 10 mins to

punch ~1000 holes in a ~19 mm diameter area (S1 and S2 Movies). Specifically, we used a PDMS sheet with 250 μm thickness. This is because thinner PDMS was too fragile to handle, while punching on a thicker PDMS created holes with cross slopes [34]. We have found that 250 μm is the optimal thickness capable of achieving relatively flat channel cross-sections (Fig 1B). The laser cutting characteristics are not affected by the PDMS base/curing ratio. We did not observe any significant difference between the 5:1 and 10:1 mixing. It is worth mentioning that despite the fact that we fabricated the 250 μm PDMS sheets using a home-made metal mold (Materials and Methods and Fig 1A), these sheets are also commercially available. More-over, once fabricated, the PDMS stencil is reusable (we did not see a big difference after one year usage). Taken together, unlike most lithography patterning systems [35], which are expensive and equipment-heavy, our protocol is more accessible and cost-efficient for most biology labs.

Using the PDMS stencil with pre-punched holes, we can pattern a glass surface into various adhesion zones without physical boundaries. As shown in Fig 2A (and described in Materials and Methods), we used the PDMS stencil to coat the exposed area with extracellular matrix (we used laminin here, but our unshown data confirms other options, such as fibronectin, also work). We then peeled the stencil off and incubated the pre-covered glass with blocker BSA. To observe the boundary of high/low adhesion zones, we used a biotin labeled version of BSA, which can be specifically bound to Streptavidin later (with Alexa405 pre-labeled). Note that, although we also used plasma treatment (Materials and Methods), it was performed after plat-ing the PDMS, i.e., only treating the opened glass surface (i.e., hole-area) to better coat the extracellular matrix. This is the opposite of most lab-on-a-chip experiments, where the plasma treatment is applied to the entire glass surface [1, 24]. In the latter case, the PDMS cannot be peeled off, but ours is detachable and reusable.

To test whether cells can be confined within the high affinity zones on the patterned surface without physical boundaries, we chose to plate mouse embryonic stem cells (ESCs). This is because: 1) This cell type is very sensitive to culturing conditions [27]. If our patterned surface works for ESC, it is more likely to be universally applicable to other cell types; 2) High effec-tive-throughput of time-lapse microscopy (as discussed in the Introduction) is essential for ESC dynamics studies. First, high resolution imaging (using at least 60x objective) is necessary, because ESC constantly forms colonies, where cells are closed attached and difficult to segment into individual cells [27, 28]. However, higher resolution often results in a smaller field of view, i.e., more severe cell-losing problems. Moreover, the high heterogeneity in the stem cell population needs, not only numerous time-lapse movies, but also the entire lineage tracking of each cell colony, to study their dynamic behaviors. Taken together, we have chosen ESC as the example system to demonstrate the efficiency of our boundary-free patterning system.

We first checked whether ESCs can sense the high-/low- adhesive zones. Specifically, we incubated a large amount of (about 250k) in a 35 mm cell dish with patterned glass-bottom surface for 4 hr, then washed the extra cells away. As shown in Fig 2B, we have found that the cells have much higher affinity to the extracellular matrix coated zone and the efficiency of the patterning is high. When the surface was not covered with laminin but only patterned with BSA (such as the top right corner of the dish, Fig 2B left), no cell patterns had formed (Fig 2B middle). This observation also confirms the high specificity of ESCs' preference for the high-adhesive zones. Note that, the no-laminin coating of the top right corner was due to a weaker attachment of the PDMS stencil edges to the glass surface. We did not further optimize the stencil attachment step, as more than 95% of the ~1000 patterned zones were forming confined patterns (Fig 2B right) and several hundreds of patterned zones are already sufficient for a typi-cal time-lapse microscope experiment. Taken together, we have found that our patterning pro-tocol can achieve efficient boundary-free adhesion zones for ESCs.

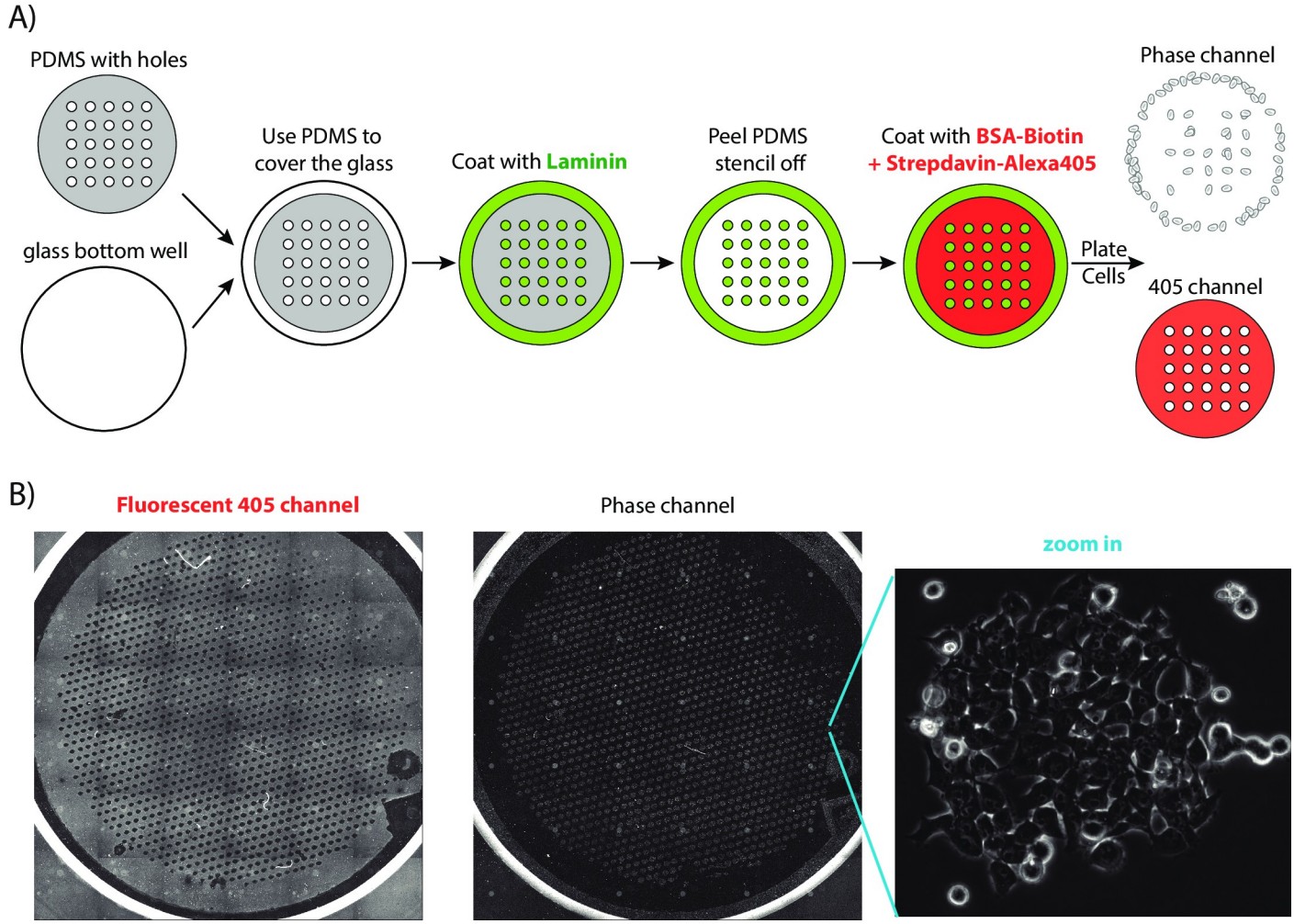

**Fig 2. ESCs stayed in the high-adhesive patterned confined area.** (A) We have used the fabricated PDMS with ~1000 holes as the stencil to coat the glass surface into high-adhesive zones (with laminin, green) and low-adhesive zones (with BSA, labeled with Alexa 405, red). (B) When plating ESCs onto the patterned surface, they formed well-shaped cell colonies in the high-adhesive zones. Worth mentioning, there are some mysterious shaded lines and colonies with high-intensity in the left and middle images. This is because, in order to cover the entire glass bottom of the 35mm culture dish, we have scanned the surface with 36 overlapping images then pieced them up. The uneven high-intensity and shade were generated due to the overlapping.

We next tested live cell growth, verifying whether ESCs can be restricted in the confined area for longer times. Specifically, we incubated low cell density of ESCs (with 1–2 cells per confined area) at day 1 and let them grow in the standard culture condition for 4 days. As shown in Fig 3, even if the cells were initially located on the edge of the confined area, they still grew into a colony within the confined area. Some cells did not grow into colonies, but this is because ESCs do not thrive in low density, regardless of the surface patterning. Remarkably, as the pattern is boundary-free, once the cell colony gets bigger, the colony can expand out (still centered around the confined area) (Fig 3), which is distinct from previous PDMS patterning systems with boundaries [24, 35].

We then performed a real time-lapse movie with the patterned system. First, we fluorescently labeled the ESC nucleus with CFP (Materials and Methods, Fig 4A) and verified its pre-embedded GFP-brachyury reporter, which indicates the transition from cells' pluripotent state to mesoderm state (data not shown). The CFP labeling, as well as the heterogeneity of CFP signals between cells, can help simplify the cell segmentation and tracking processes, as the signal

## Fluorescent 405 channel + Phase channel

**Fig 3. ESCs grew 4 days in the patterned surface.** Cells preferred growing in the confined high-adhesive (not red) area, even if they were initially located on the edge. The pattern is boundary-free, so cells started expanding out at day 4. The red background is the fluorescent 405 channel, i.e., the low-adhesive zone. The uneven shading is due to the scanned image overlapping (similar as in Fig 2B).

is more distinguishable especially compared to the phase imaging (Fig 4B). A typical 4-days recording is shown in Fig 4C and S3 Movie. Similar as in Fig 3, we have found that cells preferred growing in the high-adhesive patterned zone. Once the confined area was filled, cells expanded out.

We have further checked whether ESCs maintain their phenotype with the patterned system. First, we have observed the positive brachyury signal with high heterogeneity (as shown in Fig 4D) in 40 out 47 recorded ESC colonies at day 4, which is also consistent with previous mesoderm induction studies [36]. Second, as a comparison, we performed the no-pattern (with uniform laminin coating, without pattern or BSA) time-lapse movie (example in S4 Movie). We have found cells maintain similar motions under both conditions, and their division times are comparable too: $14.25 \pm 1.64$ hr under no-pattern condition, and $14.75 \pm 2.51$ hr under patterned system. These values are also consistent to previous publications [36]. These results together confirm that ESCs, despite their sensitivity to culturing conditions, have mostly maintained their physiology on our boundary-free patterned surface.

Finally, we have quantified how much the patterned system can improve the effective-throughput of time-lapse microscopy experiments. First, we can track more stage positions with the patterned surface. The time interval of time-lapse movie frames is often set to be 10–20 mins, because any time longer causes difficulty in cell-identification and cell-tracking due to the mobility of most cultured cells. When cells are plated randomly, the recorded positions are manually picked (a laborious process, taking hours) and mostly far apart. As both piezo-stage moving and auto-focus processes have constrained speed, requiring about 3–10 seconds to transit from one position to another, it often takes up to half of the time in the limited time intervals (the other half is for the real imaging process). Now, with the patterned surface, both position-picking and position-transit are simplified (Fig 5A), which saves half of the transit time and enables recording ~1.5 times more positions per movie. For instance, with 15mins time-interval ESC movie tracking, we have achieved picking 119 positions in total. Second, the patterned surface solves the cell losing problem and has a significantly higher chance to record the entire dynamic lineage of an ESC colony. For instance, in a typical experiment, where we tracked 47 surviving ESC colonies in 4 days, 34 of them were recorded with a full lineage (i.e.,

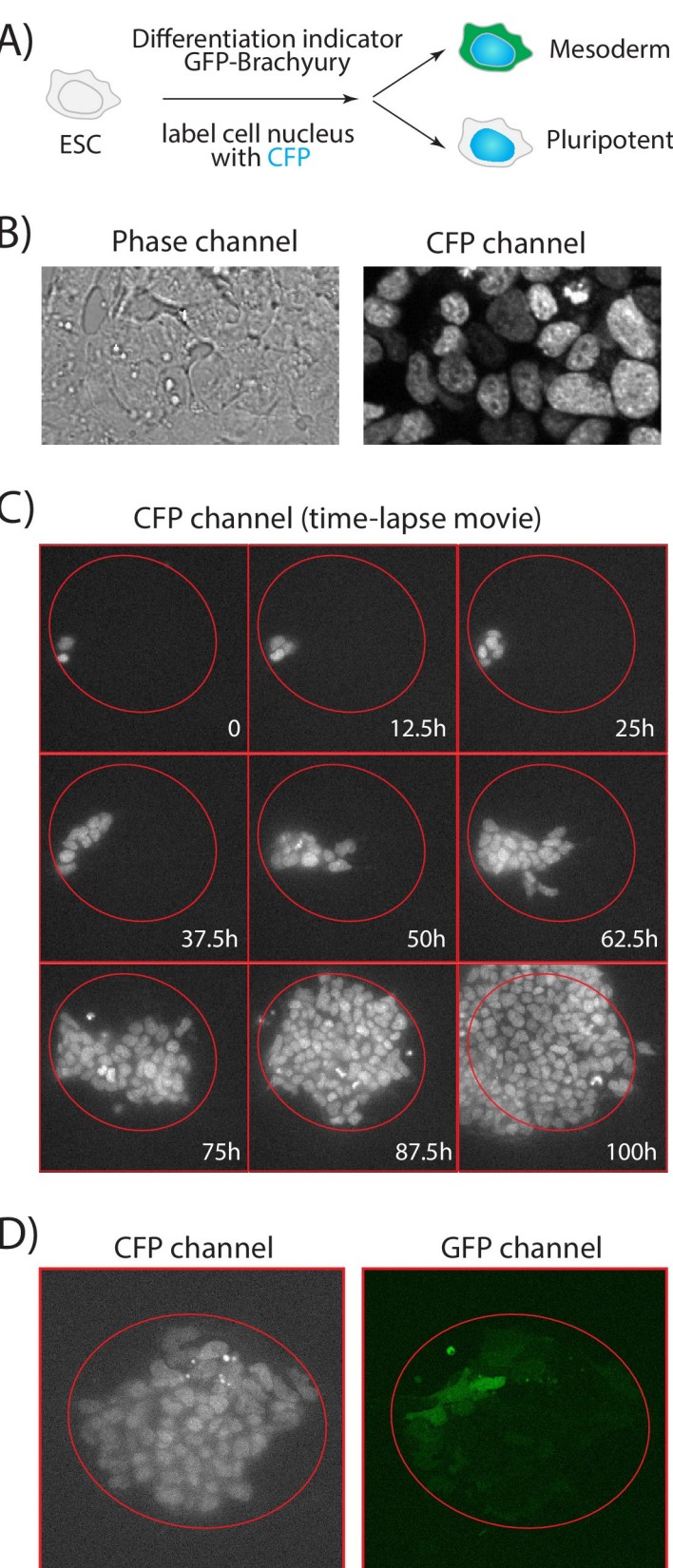

**Fig 4. ESCs maintained their phenotype on the patterned surface in a time-lapse microscopy.** A) We have labeled the ESC nucleus with CFP. The ESC has a pre-embedded differentiation indicator GFP-Brachyury. B) CFP signal has high heterogeneity in the ESC population, which helps the cell segmentation and lineage tracking. C) 9 snapshots from S3 Movie, which is a 4-days time-lapse movie of ESC on the patterned surface. The red circles indicate the confined high-adhesive area. D) A snapshot of time-lapse movie shows the ESC colony has diverse brachyury signals in the population.

no cell moved out of field of view, until the colony became too big and was squeezed out. By entire lineage, we define it as at least 90% of the colony being captured throughout the 4 days, like in Fig 4C and S3 Movie). In the other 13 positions, at least half of the colonies were captured. Together, 100% positions have captured >50% of the cell lineage and about 72.3% positions obtained the entire lineage.

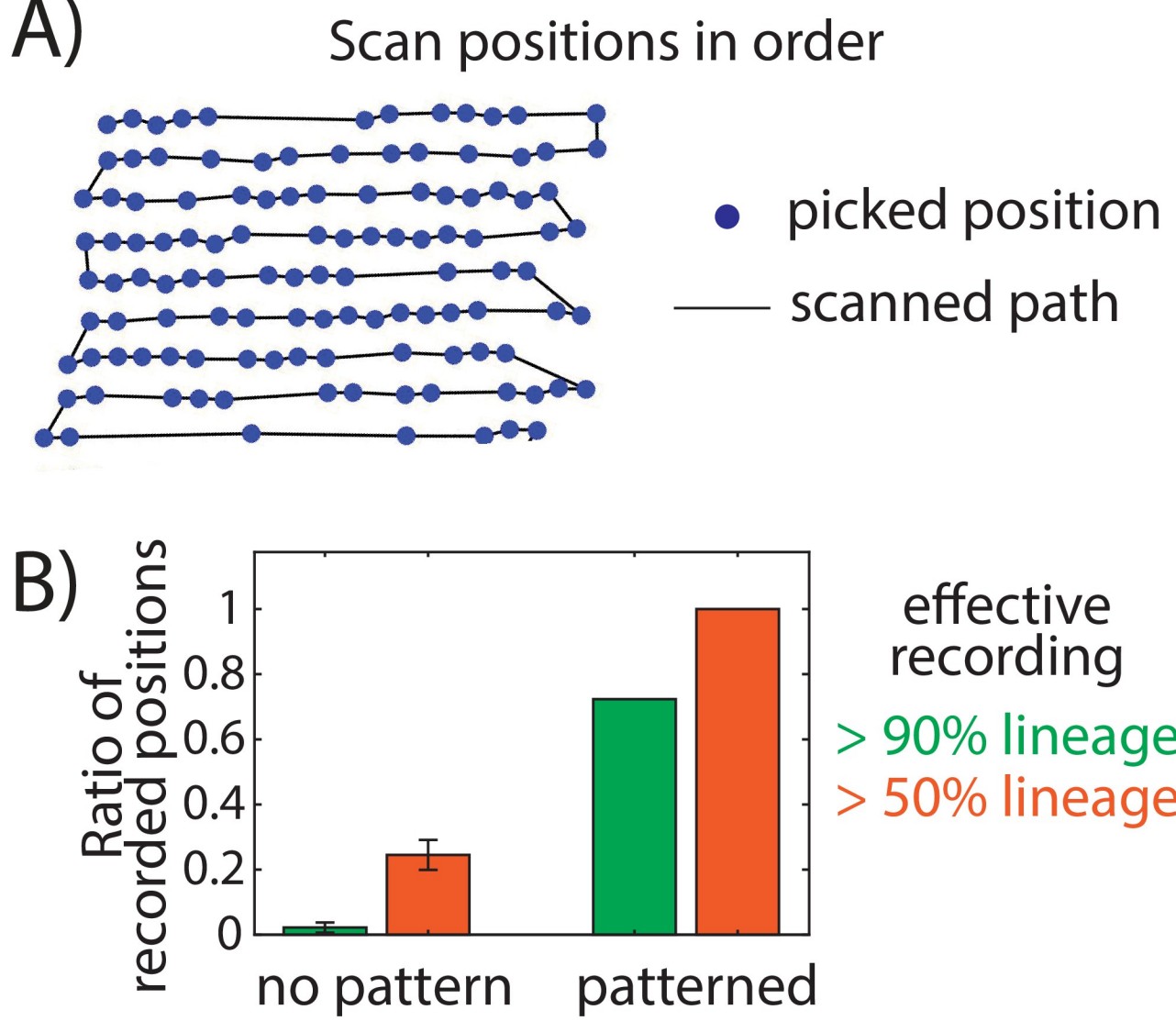

**Fig 5. The patterned surface improves the effective-throughput of time-lapse microscopy experiments.** A) Microscopy has scanned the stage positions in order. B) The majority positions of a patterned surface can record the full lineage of a cell colony, with a ratio significantly higher than the no-pattern case.

To further quantify the improvement of the effective-throughput of time-lapse microscopy experiments, we need to compare the lineage recording rate of ESCs on a patterned surface versus the no-pattern case. However, under no-pattern system, we can barely catch even a single entire lineage colony in an experiment (e.g., we only obtained one position, out of 79 tracked positions, with 50% lineage captured). Thus, it is impractical to collect sufficient data for statistics. We therefore performed a mathematical simulation to estimate the no-pattern situation, with both synchronized cell cycle and continuous stochastic growth model (more details in Materials and Methods). A typical cell-losing case is in S5 Movie. By simulating 100,000 movies, we found 24.5% ± 4.6% positions have captured >50% lineages, but only 2.2% ± 1.6% positions have >90% lineage under the synchronized cell cycle mode (Fig 5B), and similarly, 22.9% ± 4.3% positions have captured >50% lineages, but only 3.6% ± 2.0% positions have >90% lineage under the continuous stochastic growth mode. Taken together, for experiments that require full lineage tracking (like ESC differentiation studies with high heterogeneity [36, 37]), our patterned surface can help improve >30 times effective-throughput of a time-lapse microscopy experiment.

It is worth mentioning that we have focused on culturing ESCs on the patterned surface in this work. Further optimization shall be tested for different cell types, especially about the specific choices of different blockers and extracellular matrices. For instance, we have found the protocol working less efficient for C2C12, a myoblast cell line, but mixing Pluronic F-127 with BSA helped. In addition, the developed method has a limitation to cell types that have preference to different surface affinities and will not be applicable to cells capable of growing unbiasedly on either high-affinity coated, low-affinity coated, or uncoated surface.

## Conclusion

We have demonstrated a cheap and boundary-free surface patterning protocol that is adaptable to usage by a wide community (especially most standard biology labs without easy access to lithography). The patterned surface has reliable bio-compatibility, as even a sensitive cell type like the ESC maintained its phenotype during a 4-days movie recording. This system significantly reduces the cell-losing problem during time-lapse microscopy movies and improves the effective-throughput per experiment by an order of magnitude. Further studies of alternative extracellular matrix and blocker options will enable this patterning protocol applicable to other cell types and cell growth conditions.

## Supporting information

**S1 Movie. A movie showing how laser cutting machine penetrates the PDMS stencils using the raster scan mode.**
(MOV)

**S2 Movie. A movie showing how laser cutting machine outlines the PDMS stencils using the vector mode.**
(MP4)

**S3 Movie. A typical 4-days time-lapse movie with the patterned system.** We have found that cells preferred growing in the high-adhesive zone (circled). As the pattern is boundary-free, cells can expand out once the confined area was filled (at around 90 hr in the movie).
(AVI)

**S4 Movie. A typical time-lapse movie without any patterned system.** We can barely catch a single entire lineage colony. In the shown move, we lose the colony within 2 days.
(AVI)

**S5 Movie. A simulation of colony growth with the synchronized cell cycle model.**
(MP4)

## Author Contributions

**Conceptualization:** Fangyuan Ding.

**Data curation:** Guohao Liang, Hong Yin, Jun Allard, Fangyuan Ding.

**Formal analysis:** Fangyuan Ding.

**Funding acquisition:** Fangyuan Ding.

**Methodology:** Guohao Liang, Hong Yin, Fangyuan Ding.

**Project administration:** Fangyuan Ding.

**Resources:** Fangyuan Ding.

**Software:** Jun Allard.

**Supervision:** Fangyuan Ding.

**Visualization:** Fangyuan Ding.

**Writing – original draft:** Guohao Liang, Hong Yin, Jun Allard, Fangyuan Ding.

**Writing – review & editing:** Guohao Liang, Hong Yin, Jun Allard, Fangyuan Ding.

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
