## [Decision Letter · Decision Letter 0]

28 Jun 2022

PONE-D-22-12330Cost-efficient boundary-free surface patterning achieves high effective-throughput of time-lapse microscopy experimentsPLOS ONE

Dear Dr. Ding,

Thank you for submitting your manuscript to PLOS ONE. After careful consideration, we feel that it has merit but does not fully meet PLOS ONE’s publication criteria as it currently stands. Therefore, we invite you to submit a revised version of the manuscript that addresses the points raised during the review process.

I recommend you to add more (control) experiments to demonstrate the advantages of your method over existing ones. Different cell lines may be helpful.

We look forward to receiving your revised manuscript.

Kind regards,

Kun Chen, Ph.D

Academic Editor

PLOS ONE

Journal Requirements:

Reviewers' comments:

Reviewer's Responses to Questions

**Comments to the Author**

1. Is the manuscript technically sound, and do the data support the conclusions?

Reviewer #1: Yes

Reviewer #2: No

2. Has the statistical analysis been performed appropriately and rigorously? 

Reviewer #1: Yes

Reviewer #2: Yes

3. Have the authors made all data underlying the findings in their manuscript fully available?

Reviewer #1: Yes

Reviewer #2: No

4. Is the manuscript presented in an intelligible fashion and written in standard English?

Reviewer #1: Yes

Reviewer #2: No

5. Review Comments to the Author

Reviewer #1: In this work, Liang et al. developed one simple method for high-throughput of time-lapse live cell experimental studies. In this method, the cells will be confined to high-adhesive zones through a cheap and boundary-free surface patterning protocol. This simple and cost-efficient method solved the cell-losing problem during time lapse microscopy movies and improved the effective-throughput experiments. Furthermore, the authors demonstrated that the ESC cell maintained their phenotype in a 4-day long experiment. Overall, this is a timely piece of study for live cell imagining studies, as cell loss is a big problem for long-time experiments. Here is are some comments:

1. Is it easier to do a simple experiment for the no-pattern situation and calculate the lineage recording rate?

2. The authors need to detail the imaging analysis for lineage tracking.

3. The authors need to detail the simulation method for estimating the lineage recording rate in Fig.

4. Fig. 3, why does the image on day 3 have a much brighter background than the others?

Reviewer #2: In this manuscript, Liang et al. describe a method that restricts cell migration to pre-defined boundary-free areas via differential cell-ECM affinities. The authors claim that this method is low-cost and that it significantly improves the throughput of time-lapse microscopy experiments by preventing cells from migrating out of the fields of view. The major caveat of this manuscript is that the authors do not provide sufficient benchmarks to demonstrate the advantages of their methods over the state-of-the-arts. Therefore, it remains difficult to evaluate whether their method provides a meaningful contribution to the community. Specifically, the authors should answer the following questions.

1. Does laminin and BSA coating alter cell phenotypes?

Factors such as ECM stiffnesses, local cell density, and the geometry of the culturing environment are known to regulate cell phenotypes. Therefore, a central task is to demonstrate that the coating does not significantly alter the cell phenotypes. While the authors demonstrated that the brachyury signals in ESCs on Day 4 matched the previous studies, this only reflects one aspect of cell phenotype in a highly specific experimental condition. The authors could perform parallel experiments in coated and uncoated plates and examine whether cell proliferation (e.g., quantify the intermitotic times by time-lapse microscopy or stain pRb S807/811 by IF) and the expression levels of stem-cell marker genes (by IF) are consistent. The authors should also consider comparing cells in the high-affinity areas, the low-affinity areas, and the transitory areas to demonstrate that the abrupt change in the cell-ECM affinity does not alter the cell phenotypes. Finally, benchmarking on other cell lines, such as cancer cells where ECM stiffnesses are known to regulate their therapeutic responses, will be useful. While it is understandable that no methodology is perfect, demonstrating the scope of applicability is necessary.

2. Does this method provide similar performance to other state-of-the-art methods, such as photolithography, despite being cheaper?

The authors could perform similar benchmarks described above in laminin/BSA-coated plates and photolithography-modified plates.

3. Does this method allow one to track more cells throughout the imaging period?

In addition to the cells migrating out of the fields of view, the throughput of time-lapse microscopy experiments is governed by the accuracy of automatic cell tracking. As cells become more crowded with this method, the accuracy of segmentation (detecting cell nuclei) and track linking (mapping cell nuclei between images and detecting mitosis, etc) could decrease. It is thus necessary to track the cells cultured in coated and uncoated plates with standard automatic cell-tracking algorithms and examine the number of cells tracked throughout the imaging period and the accuracy of tracked cell lineages. If this method is only aimed at manual cell tracking, it should be clearly described.

Minor Comments

1. The authors provide many technical details in the “Results and Discussion” section, such that it becomes too wordy. The authors could consider summarizing their findings in the main text and additionally providing a protocol with these technical details in the supplementary materials.

2. The authors should proofread their manuscripts to match the publication standard. For example, “250um” should be “250 μm” (notice the space between the number and the unit), and in the text “For the 35cm glass-bottom dish”, “35cm” should be “35 mm.”

3 (Optional). The authors used a simple numeric simulation to demonstrate the improvement of the experimental throughputs. While it is effective, the authors could easily make the simulation more realistic, such as by simulating the cell division with the Gillespie algorithm. Alternatively, the authors could formulate their problem by calculating the first-passage time of an unbiased random walk, for which the closed-form solution exists.

6. PLOS authors have the option to publish the peer review history of their article (what does this mean?). If published, this will include your full peer review and any attached files.

Reviewer #1: No

Reviewer #2: **Yes: **Chengzhe Tian

---

## [Author Response · Author response to Decision Letter 0]

13 Sep 2022

Please find details in the "Response to Reviewers" document.

---

## [Decision Letter · Decision Letter 1]

26 Sep 2022

Cost-efficient boundary-free surface patterning achieves high effective-throughput of time-lapse microscopy experiments

PONE-D-22-12330R1

Dear Dr. Ding,

We’re pleased to inform you that your manuscript has been judged scientifically suitable for publication and will be formally accepted for publication once it meets all outstanding technical requirements.

I recommend you to consider the minor comments about the language of the manuscript from one of our reviewer before the final publication.

Kind regards,

Kun Chen, Ph.D

Academic Editor

PLOS ONE

Additional Editor Comments (optional):

Reviewers' comments:

Reviewer's Responses to Questions

**Comments to the Author**

1. If the authors have adequately addressed your comments raised in a previous round of review and you feel that this manuscript is now acceptable for publication, you may indicate that here to bypass the “Comments to the Author” section, enter your conflict of interest statement in the “Confidential to Editor” section, and submit your "Accept" recommendation.

Reviewer #1: All comments have been addressed

Reviewer #2: (No Response)

2. Is the manuscript technically sound, and do the data support the conclusions?

Reviewer #1: Yes

Reviewer #2: Yes

3. Has the statistical analysis been performed appropriately and rigorously? 

Reviewer #1: Yes

Reviewer #2: Yes

4. Have the authors made all data underlying the findings in their manuscript fully available?

Reviewer #1: Yes

Reviewer #2: Yes

5. Is the manuscript presented in an intelligible fashion and written in standard English?

Reviewer #1: Yes

Reviewer #2: No

6. Review Comments to the Author

Reviewer #1: Thanks the authors for the comprehensive and thoughtful responses. The authors have adequately addressed all my comments. I congratulate them on this great work!

Reviewer #2: The authors have significantly improved the manuscript according to the comments. Scientifically, I believe that this manuscript has reached the publication standard of PLoS ONE. However, since PLoS ONE does not provide copyediting, the authors should consider further improving the language of the manuscript to attract greater audience, and letting a native speaker proofread the manuscript could help. Therefore, I suggest a decision of either acceptance or minor revision (specifically for the language).

For example, the second sentence of the Abstract looks weird. Since the proposed method does not address the "laborious" part, mentioning this in the abstract is not necessary. The structure could be: the first part introduces the definition of the cell-losing problem (the content in the parenthesis), and the second part describes the consequences: very few cells tracked in long-term recordings.

The first paragraph of "Results and Discussion" still sounds too wordy. For example, the sentence "Moreover, once fabricated, the PDMS stencil is reusable..." could be a single short sentence without parenthesis.

Additionally, some numbers and units are still incorrect. Some numbers and units are still incorrect. For example, in "Analysis of colony movement", "15um" should be "15 μm." There are at least 10 such instances in the manuscript.

Finally, the authors could mention that another caveat of the PDMS-based physical confinement approach: PDMS will absorb small hydrophobic molecules (https://royalsocietypublishing.org/doi/10.1098/rsob.160156), such that it could not be used in many biological research fields (e.g., cell cycle).

7. PLOS authors have the option to publish the peer review history of their article (what does this mean?). If published, this will include your full peer review and any attached files.

Reviewer #1: No

Reviewer #2: **Yes: **Chengzhe Tian

---

## [Editor Report · Acceptance letter]

18 Oct 2022

PONE-D-22-12330R1 

Cost-efficient boundary-free surface patterning achieves high effective-throughput of time-lapse microscopy experiments 

Dear Dr. Ding:

I'm pleased to inform you that your manuscript has been deemed suitable for publication in PLOS ONE. Congratulations! Your manuscript is now with our production department. 

Kind regards, 

on behalf of

Dr. Kun Chen 

Academic Editor

PLOS ONE